# Classification of Higher-Order Thinking Skills of the Teachers Based on Institution, Seniority, and Branch with Discriminant Analysis

**DOI:** 10.3390/bs14080626

**Published:** 2024-07-23

**Authors:** Özlem Ulu-Kalin, Hatice Kumandaş-Öztürk

**Affiliations:** Faculty of Education, Artvin Coruh University, 08000 Artvin, Türkiye; haticekumandas@yahoo.com

**Keywords:** teacher, thinking skill, discriminant analysis

## Abstract

The present study aimed to determine the variations between higher-order thinking skills (critical, creative, and empathetic thinking) of the teachers based on institution, branch, and professional seniority. The study data were collected from 345 teachers with the Critical Thinking Tendency Scale, Marmara Creative Thinking Aptitudes Scale, and Basic Empathy Scale, and confirmatory factor analysis was conducted on the data. The data were analyzed with discriminant analysis. The study findings demonstrated that there were significant differences between the critical, creative, and empathetic thinking skills of the teachers based on the institution of employment, their seniority, and their branch. The analysis of the discriminant functions revealed that the most discriminatory variables were empathy based on the institution and creative thinking based on seniority and branch. It was determined that the teachers employed in pre- and primary schools with less than 15 years of seniority scored higher in critical thinking skills, and teachers who were employed in middle schools with 16 years or more seniority scored higher in creative thinking skills. Foreign language teachers scored higher in both thinking skills. Finally, the mean empathy skill scores of the teachers employed in pre- and primary schools with 15 years or less seniority in the common branch were higher.

## 1. Introduction

Thinking is one of the most distinctive features that distinguish humans from other living beings. Thinking skill is the ability of individuals to analyze, evaluate, synthesize, and use information. Thinking skills are necessary for effective data processing and the production of creative problem solutions. Contemporary education systems aim for the acquisition of advanced thinking skills from an early age, in line with contemporary requirements. Higher-order thinking skills entail the cognitive capacity that goes beyond the given information to classify, infer, generalize, and solve problems in complex situations [1]. Higher-order thinking skills are an umbrella phenomenon that includes several skills, such as critical, analytical, creative, reflective thinking, decision-making, and problem-solving [2]. Thinking skills play a key role in the daily, business, and academic lives of individuals and in personal development and allow individuals to make more conscious, flexible, and effective decisions [3]. 

High thinking skills are extremely important among teachers to improve the quality of education and student development [4]. High thinking skills play a critical role in education and student development [5]. Teachers should acquire thinking skills and implement them in their daily lives. Only then could they conduct activities with their students to improve the skills of the latter [6].

Critical, creative, and empathetic thinking skills would assist the teachers in solving problems inside and outside the classroom, supporting the social and emotional development of students, instructing interesting classes, and developing innovative instruction methods [7,8].

In general, it is described as a set of complex cognitive skills involved in critical thinking, problem-solving, and intellectual thinking [9]. Currently, critical thinking skills are included in curricula and evaluation guides in several countries, and efforts have been made to disseminate this skill in society [10,11,12]. Thus, the acquisition of these skills is a requirement for the training of a generation that can keep up with the requirements of the age [13]. Critical thinking was described as a higher thinking level that aims at perfect thinking [14]. To achieve this goal, existing misinformation should be eliminated, and access to accurate information should be promoted [15]. A critical perspective should be developed through analysis, synthesis, and deduction based on accurate data.

Certain standards are required for critical thinking. These standards are necessary for the logical analysis of an idea. Nosich proposed seven critical thinking standards: clarity, accuracy, significance, adequacy, depth, breadth, and precision [16]. Furthermore, the following were determined as the critical thinking standards: logical, logical-rational, consistent, falsifiable, testable, well-organized, reliable, and effective. McKnown grouped critical thinking components into two categories: determination of the reasons behind thinking and spending a certain effort to think [17].

Critical thinking is comparable to reflective thinking, which allows the individual to focus on when they decide what they believe in or do [18]. Critical thinking is not an innate trait but a system that can be taught and instructed. Critical thinking leads to better learning [19] and higher analysis skills [20]. Thus, teachers should have critical thinking skills. Certain studies investigated the thinking skills of the teachers [21,22,23,24].

In addition to critical thinking, creative thinking, a higher-order thinking skill, is a significant dimension of learning. Creative thinking includes behaviors that allow students to develop original and imaginary products and make judgments about the value of their work [25]. There are several descriptions of creative thinking in the literature. Creative thinking is the process of internalizing the problems encountered by individuals by sensing these problems, finding various predictions and suggestions about the solution to these problems by determining the causality between related events, producing alternative solutions, comparing the outcomes, and developing various variable products. It entails going beyond the ordinary and developing an original and innovative perspective [26]. Creative thinking is the ability to develop new ideas about an emerging requirement [27]. The authoritarian attitudes of the teachers could negatively affect the creative thinking skills of children [28].

According to Senemoğlu, creativity is the ability to find new solutions to unsolved problems, approach them with new ideas, and come up with new inventions [29]. Fisher classified the stages of creativity as stimulation, discovery, planning, activity, and review [30]. Education and teacher and student material should be revised based on creative thinking requirements to train qualified individuals. Learning activities that aim to improve creativity require learners to assume the role of problem solvers and communicators rather than passive information recipients [31].

Students who are instructed in creative classroom environments exhibit creative, problem-solving, brave, self-confident, and open-to-criticism behavior [32]. Teachers who instruct these behaviors should possess creative thinking skills. Previous studies were conducted on the creative thinking aptitudes of the teachers [33,34]. Rıza reported student problems that prevented the development of creative thinking skills and emphasized the role of emotional, cultural, learned, and loaded curricula [35]. Emir and Bahar listed teacher traits that hindered creativity and reported that teachers were oppressive and authoritarian, did not value student opinion, were anti-democratic, and were not innovative [36].

According to Aykaç and Adıgüzel, instruction of the courses with methods that promote creative thinking would contribute to better learning the objectives, understanding the subject, and interest in the course. Furthermore, it would raise the student’s curiosity about the course [37].

Empathy is a response parallel to the emotional and cognitive state of another. Empathy plays a key role in human life due to its ability to change human behavior [38]. Teachers’ empathetic thinking skills help establish a deeper bond with the students and understand their emotional needs.

Empathy was described by Dökmen as the process of individuals putting themselves in the shoes of others and having the ability to look at events from others’ perspectives, correct their comprehension of others’ emotions and thoughts, and communicate the above to these individuals [39]. Effective teaching, communication, and classroom management skills of teachers are associated with empathy skills [40]. It was evidenced that individuals with high empathy skills could establish close relationships and easily communicate with others [41].

Teacher competency is critical to high-level thinking skills, especially critical, creative, and empathetic thinking skills. These skills would improve the effectiveness and permanence of education [42,43,44,45,46]. Several studies were conducted on the differences between these skills based on institution, branch, and seniority [47,48] or the impact of other variables on them [49,50]. However, no previous study has classified the impact of teacher skills on these variables. Thus, the present study aimed to determine the differences between the thinking skills (critical, creative, and empathetic) of the teachers based on institution, branch, and professional seniority variables to fill this gap in the literature.

## 2. Material and Method

### 2.1. The Research Model

This study was conducted using the general survey model. According to Karasar, the survey model aims to describe a past or present case where the researcher does not change or affect the conditions of the individuals or objects [51]. In surveys, data collection and analysis are important to determine the facts about past events and make judgments [52]. The present study adopted a correlational research model, where the correlations between two or more variables were investigated without interference.

### 2.2. The Study Group

The study group included 354 teachers employed in public primary (preschool, primary, and middle schools) and secondary education institutions in Turkey. In the analysis, 9 individuals were determined to be outliers, and these individuals were excluded from the study data. This study was conducted with 345 teachers. The institution of employment, professional seniority, branch, and gender of the participants are presented in Table 1.

Teacher branches were categorized based on their undergraduate education. Teachers trained in mathematics and science were included in the numerical category; teachers trained in mathematics and Turkish were included in the general category; teachers trained in Turkish and social sciences were included in the verbal category; teachers trained in music, art, and sports were included in the talent category; and teachers trained in foreign languages were included in the foreign languages category.

### 2.3. Data Collection Instruments

The following data collection instruments were employed:

*Personal Data Form*: It was developed by the authors. The survey included structured questions to collect personal information (gender, seniority, branch, etc.) about the participants.

*Critical Thinking Aptitude Scale*: the scale, adapted into the Turkish language by Ertaş, includes 25 items [53]. The internal consistency coefficient of the overall scale was calculated as 0.972, the participation dimension coefficient was 0.884, the cognitive maturity dimension coefficient was 0.774, and the innovation dimension coefficient was 0.783.

*Marmara Creative Thinking Aptitudes Scale*: The scale was developed by Özgenel and Çetin and includes 25 items and 6 factors [54]. Cronbach’s alpha internal consistency coefficient for the entire scale is 0.92.

*Basic Empathy Scale*: The scale was adapted to the Turkish language by Topcu, Erdur-Baker, and Çapa-Aydın [38]. The Cronbach’s alpha coefficient of the Likert-type scale is 0.775 for the emotional dimension and 0.769 for the cognitive dimension.

Confirmatory factor analysis was conducted on all scales, and goodness of fit indices were determined to be within acceptable limits. χ^2^/df, RMSEA, and CFI were employed as statistical fit criteria for the models validated in the confirmatory factor analysis. For a good model, the χ^2^/df should be small. A ratio between 2 and 3 indicates that the fit is acceptable or good for the model [55]. Based on the goodness of fit indices, χ^2^/df (264.55/169) was 1.56 for the empathy scale, χ^2^/df = (367.05/275) = 1.33 for the critical thinking scale, and χ^2^/df = (299.09/275) = 1.08 for the creative thinking scale. These coefficients are accepted as a perfect fit in large samples [56,57]. An RMSEA less than or equal to 0.05 indicates a perfect fit, while a value greater than 0.10 indicates an unacceptable model [56]. In the present study, RMSEA was 0.08; SRMR was 0.0 for the empathy scale; RMSEA was 0.09; SRMR was 0.05 for the critical thinking scale; RMSEA was 0.05; and SRMR was 0.04 for the creative thinking scale. Comparative fit indices CFI, AGFI, and NNFI were within acceptable limits for all three measurement instruments (CFI > 0.90, AGFI > 0.95, and NNFI > 0.95).

### 2.4. Data Analysis

The study data were analyzed using discriminant analysis to accurately determine the most effective variable in the thinking skills of the teachers.

Discriminant analysis groups the variables in a certain (X) dataset into two or more categories and derives functions that would allow the optimal assignment of these units to real groups in the natural environment based on the p properties of the units [58]. This method allows the determination of the differences between two or more groups based on the discriminant variables [59].

Stevens reported that discriminant analysis could be used for two purposes [60]:To determine the key differences between the groups in MANOVATo classify group members based on scale scores.

Thus, discriminant analysis determines differences between the groups and assigns units to them. Therefore, discriminant analysis is an analytical prediction method [61].

Before discriminant analysis, the assumptions were reviewed, and the findings are summarized below.

### 2.5. Outliers

Discriminant analysis is quite sensitive to outliers. Although certain outliers could reflect the characteristics of the population, they could also exhibit deviations. In the present study, standard z values were calculated in the outliers dataset. Nine out of the calculated z values were removed from the dataset because they were between −3 and +3, and then their Mahalonobis distances were determined. These values were compared with the χ2 table at the 0.01 level. No dataset was greater than the table. Thus, 9 values were removed from the dataset, leaving 345 participant data in the dataset.

Multivariate Normal Distribution

The quantitative analysis variables (independent or predictive variables) should exhibit a multivariate normal distribution [61,62]. Thus, the normal distribution of the continuous variables was investigated with histograms and p–p graphs. The results are presented in Figure 1.
Homogeneity of the covariance matrices

This assumption was tested with the Box’s M statistics. A significant Box’s M statistic (*p* < 0.05) demonstrates non−homogeneous covariance matrices and a significant difference between the covariance matrices. A significant Box’s M statistic could indicate unequal covariance matrices, deviations from normality, or both [63]. Box’s M statistics findings for all study variables are presented in Table 2.

As seen in Table 2, Box’s m statistics were not significant for all study variables (*p* > 0.05). This finding demonstrated that the covariance matrices were homogeneous.
▪Multicollinearity

Multicollinearity was tested by analyzing the correlations between the independent study variables. The multicollinearity standard was reported as 0.80 or above in the literature [64]. In the present study, there was no multicollinearity issue since the correlations between the independent variables were not 0.80 or above. Furthermore, among the independent variables, the tolerance (1–R2) was greater than 0.20, the variance inflation factor (VIF) was less than 10, and the condition index (CI) was less than 30. Thus, there was no multicollinearity problem across the independent variables.

## 3. Results

In the discriminant analysis conducted to determine whether the institution, seniority, and branch variables had an impact on the thinking skills of the teachers, eigenvalues, variances, and canonical correlations were calculated for the discriminant functions. Thus, the group statistics calculated for the three variables are presented in Table 3, and the eigenvalues and Wilks’ Lambda statistics are presented in Table 4.

It was determined that the mean group scores of the foreign language teachers employed in pre- and primary schools and who had 15 years or less experience scored higher in critical thinking skills. The foreign language teachers employed in middle schools with a seniority of 16 years or above scored higher in creative thinking skills. Furthermore, general education teachers employed in pre- and primary schools with 15 years or less experience scored higher in empathy skills.

As seen in Table 4, all variables explained between 59% and 100% of the variation. The canonical correlation coefficients revealed that the correlations between the primary functions and the groups were higher than the other functions, and the discriminant functions were ranked based on their significance in explaining the differences between the groups. Furthermore, the chi-square coefficient demonstrated that the primary discriminant functions were significant for all variables (*p* < 0.05).

The results of the test of equality of group means are presented in Table 5.

As seen in Table 4, there were differences between the empathic thinking skill scores based on institution (F = 38.67; *p* < 0.05) and branch (F = 3.70; *p* < 0.05) and between the creative thinking skills based on seniority (F = 32.42; *p* < 0.05) and branch (F = 4.90; *p* < 0.05). There were no differences between these skill scores based on seniority (*p* > 0.05).

The structure matrix that determines the discriminant functions and the correlations between these functions and variables is presented in Table 6.

The analysis of the discriminant functions demonstrated that in the first function, the most differentiating variable for teacher skills was empathy based on the institution. It was determined that there were differences between creative thinking scores based on seniority and branch. Structure coefficients revealed that the first function exhibited a high load in critical and creative thinking skills, and the second function exhibited a high load in empathy skills based on institution.

The predicted groups per variable are presented in Table 7.

As seen in Table 7, the accuracy of the group membership classification was 47% based on the institution variable, 63% based on the seniority variable, and 29% based on the branch variable.

## 4. Discussion

The findings of the current study, conducted with the correlational survey method to determine the differences between the higher-order thinking skills (critical, creative, and empathic) of the teachers based on institution, branch, and seniority variables, were as follows:

Discriminant analysis was employed to analyze the data collected from 345 teachers using the Critical Thinking Aptitude Scale, Marmara Creative Thinking Scale, and Basic Empathy Scale data collection instruments, and confirmatory factor analysis was conducted on the data. Before the analysis, assumptions such as outliers, multivariate normal distribution, covariance matrices, and multicollinearity were tested.

As seen in Table 3, the analysis revealed that the mean critical thinking skill significance of the preschool and primary school teachers (103.99), that of the teachers with 15 years or less seniority (103.18), and that of the foreign language teachers (103.71) were higher when compared to other teacher groups. Other studies reported positive correlations between primary school teachers’ critical thinking skill instruction levels and their competencies [22], and teachers’ critical thinking aptitudes were moderate [23].

In a study that surveyed 272 classroom teachers employed in Turkey, it was concluded that the teachers believed that they were completely competent in the instruction of critical thinking skills and implemented all applications [35]. Thus, the two studies reported similar findings.

In the 2013–2014 academic year, a mixed-method study collected data from 700 teachers in the quantitative dimension and 16 teachers in the qualitative dimension with the California Critical Thinking Aptitude Scale, and it was determined that the critical thinking aptitudes of the teachers were moderate [23].

It was determined that the mean creative thinking skill score of the foreign language teachers employed in middle schools with 16 or more years of experience was high. This finding was consistent with the reports by Ballı and Özgenel [65] and Türkdoğan and Özgenel [33].

Similar to the critical thinking skills, the mean empathic thinking skill scores of the teachers employed in pre- and primary schools with 15 years or less experience were high. The difference between the empathy and critical thinking skill scores was the high mean score of the general education teachers when compared to that of the foreign language teachers. This could be due to the fact that the population instructed by preschool and primary school teachers might not express themselves accurately and completely, leading to higher empathic thinking skills among these teachers.

The analysis of the discriminant functions and the correlations between these functions and variables revealed that the most discriminant variable was empathy in the first function based on the institution and creative thinking based on seniority and branch. The analysis demonstrated that the comparison of the findings was significant. The structure coefficients revealed that the first function exhibited a high load in critical and creative thinking skills, and the second function exhibited a high load in the empathy variable, based on the institution. Thus, it could be suggested that preschool and primary school teachers had higher critical and creative thinking skills, and high school teachers had higher empathy skills. The first function was more loaded based on seniority. In other words, low seniority was more effective in discriminating against these variables. In the analysis of the structure coefficients based on branch, the first function exhibited a higher coefficient in critical thinking, the second function exhibited a higher coefficient in creative thinking, and the third function exhibited a higher coefficient in empathic thinking.

The study findings demonstrated that the critical thinking and empathy skills of primary education teachers with 15 years or less experience were generally higher, and their creative thinking skills decreased with seniority. Thus, it would be beneficial to investigate the impact of the habits formed with seniority and burnout on these skills. Furthermore, the finding that the foreign language teachers scored higher in the analyzed skills was significant and should be noted. Teacher skills, their competencies in different types of intelligence, and their interests and experiences could also be investigated in future studies. Future studies should investigate whether these skills were acquired during higher education or later. If the acquisition of these skills during higher education differs based on the branch, hence the department, similar instructional methods should be adopted in all college departments.

## Figures and Tables

**Figure 1 behavsci-14-00626-f001:**
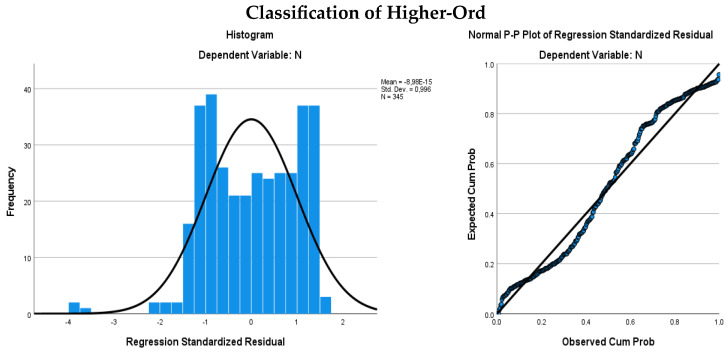
Multivariate Normal Distribution.

**Table 1 behavsci-14-00626-t001:** Participant demographics.

		Frequency	Percentage
Institution	Preschool and primary school	68	19.7
	Middle school	138	40
	High school	139	40.3
Seniority	0–15 years	212	61.4
	16 years or higher	133	38.6
Branch	Numerical	79	22.9
	Common	69	20
	Verbal	132	38.3
	Skill	37	10.7
	Foreign language	28	8.1
Gender	Female	221	64.1
	Male	124	35.9

**Table 2 behavsci-14-00626-t002:** Box’s M test findings.

		Institution	Seniority	Branch
Box’s M		14.88	11.64	33.98
F	Approx.	1.22	1.89	1.38
	df1	12.00	6.00	24.00
	df2	251,632.99	129,933.73	68,116.90
	Sig.	0.26	0.08	0.10

**Table 3 behavsci-14-00626-t003:** Group statistics.

			Mean	Std. Deviati.	N
Institution	Preschool/Primary School	critical_think	103.99	10.86	68
		creat_think	100.85	10.52	68
		Empathy	63.59	4.43	68
	Middle School	critical_think	103.14	9.94	138
		creat_think	102.65	10.50	138
		Empathy	60.89	3.54	138
	High School	critical_think	102.53	8.81	139
		creat_think	102.10	10.55	139
		Empathy	58.31	4.55	139
Seniority	0–15 years	critical_think	103.18	10.05	212
		creat_think	99.63	9.26	212
		Empathy	60.60	4.38	212
	16 years or above	critical_think	102.87	9.10	133
		creat_think	105.97	11.23	133
		Empathy	60.03	4.88	133
Branch	Numerical	critical_think	102.92	10.63	79
		creat_think	103.04	13.66	79
		Empathy	59.76	4.61	79
	Common	critical_think	103.09	9.32	69
		creat_think	100.64	9.00	69
		Empathy	62.12	5.28	69
	Verbal	critical_think	103.17	9.22	132
		creat_think	100.51	8.48	132
		Empathy	59.77	4.15	132
	Skill	critical_think	102.43	8.47	37
		creat_think	102.73	7.34	37
		Empathy	60.16	4.65	37
	Foreign language	critical_think	103.71	11.81	28
		creat_think	109.43	12.95	28
		Empathy	61.04	3.53	28

**Table 4 behavsci-14-00626-t004:** Canonical Discriminant Functions.

	Function	Eigenvalue	% of Variance	Canonical Correlation	Wilks’ Lambda	Chi-Square	df	Sig.
Institution	1	0.23	98.60	0.44	0.81	72.72	6.00	0.00 *
	2	0.00	1.40	0.06	1.00	1.13	2.00	0.57
Seniority	1	0.12	100.00	0.32	0.89	38.15	3.00	0.00 *
Branch	1	0.06	58.80	0.24	0.90	35.70	12.00	0.00 *
	2	0.04	40.40	0.20	0.96	14.79	6.00	0.02
	3	0.00	0.70	0.03	1.00	0.27	2.00	0.87

* *p* < 0.01.

**Table 5 behavsci-14-00626-t005:** Tests of Equality of Group Means.

		Wilks’ Lambda	F	df1	df2	Sig.
Institution	critical_think	1.00	0.52	2	342	0.60
	creat_think	1.00	0.67	2	342	0.51
	Empathy	0.82	38.67	2	342	0.00 *
Seniority	critical_think	1.00	0.08	1	343	0.78
	creat_think	0.91	32.42	1	343	0.00 *
	Empathy	1.00	1.28	1	343	0.26
Branch	critical_think	1.00	0.08	4	340	0.99
	creat_think	0.95	4.90	4	340	0.00 *
	Empathy	0.96	3.70	4	340	0.01 *

* *p* < 0.01.

**Table 6 behavsci-14-00626-t006:** Standardized Canonical Discriminant Function Coefficients and Structure Matrix.

	Discriminant Function Coefficients				Structure Matrix		
		1	2		1	2	
Institution	critical_think	0.12	−0.43		0.98	0.17	
	creat_think	−0.18	1.04		0.11	−0.07	
	Empathy	0.99	0.12		−0.07	0.91	
Seniority	critical_think	−0.40			0.89		
	creat_think	1.05			−0.18		
	Empathy	−0.23			−0.04		
Branch	critical_think	−0.32	−0.05	1.01	0.95	0.01	0.31
	creat_think	1.06	−0.03	−0.04	0.03	0.99	0.06
	Empathy	−0.00	1.00	−0.03	0.04	0.03	0.99

**Table 7 behavsci-14-00626-t007:** Predicted Group Memberships.

Predicted Group Membership						
		Pre- and Primary School	Middle School	High School		
Institution %	Pre- and primary school	61.8	14.7	23.5		
	Middle School	34.8	29.0	36.2		
	High School	19.4	22.3	58.3		
47.2% of the original classification was accurate.						
Seniority %		0–15 year	16 years or above			
	0–15 years	70.8	29.2			
	16 years or above	48.1	51.9			
63.5% of the original classification was accurate.						
		Numerical	Common	Verbal	Skill	For. Lang.
Branch %	Numerical	3.8	22.8	35.4	1.3	36.7
	Common	8.7	46.4	23.2	7.2	14.5
	Verbal	11.4	31.1	34.1	8.3	15.2
	Skill	8.1	35.1	21.6	10.8	24.3
	For. Lang.	3.6	17.9	17.9	7.1	53.6
28.7% of the original classification was accurate.						

## Data Availability

If the authors are contacted, the data from the study can be accessed at “URL https://docs.google.com/forms/d/12yw-ze83PHCOsL0ViA9JUT2W7Q-7vRPe4oeaGtEZ86c/edit (accessed on 19 July 2024)”.

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
