# Peer review of "Classification of Higher-Order Thinking Skills of the Teachers Based on Institution, Seniority, and Branch with Discriminant Analysis"

_behavsci, 2024, doi:10.3390/bs14080626_

Round 1

Reviewer 1 Report

Comments and Suggestions for Authors

This study is interesting, but more work needs to be done to help the audience and the field understand the value of the findings, why the findings are important, what the implications are and how could they be approached or drive future research. This is touched on in the discussion section, but highlighting these points early on in the paper will help the audience connect with the findings. I’ve included some suggested revisions.

Early in the introduction “higher order thinking skills” (used in the title) should be defined and cited before elaborating on the individual sills.

Many claims are made early in the paper about the importance of HOTs and these should be cited as well. This helps the audience understand that the importance of these skills is established beyond the authors opinions.

For Section 1 I would encourage the authors to connect their overview of the skills to current, highly cited literature which creates a common understanding in the field. This will enhance the chances that the field will find it relevant.  Include a commonly used definition of critical thinking and creative thinking if possible and relate those definitions to the chosen scales used for the study.

Line 62, briefly summarize the findings of the 4 studies, it is unclear why they are included.

Line 119 – Please elaborate on the ‘branches’, it is, for example, unclear what a ‘common’ branch might be, or a ‘skill’ branch. This will help readers to generalize the findings.

Line 254 -  “the mean critical thinking skill score of the foreign language teachers employed in pre- and primary schools, with 15 years or less experience was higher” – please be explicit about what these scores were ‘higher’ than. I’m not sure if there is a statement missing with the citation [11] and there appears to be some missing information here.

Line 255 – it is unclear what the reference to Polat is referencing, perhaps this is part of the missing information?

This study was interesting, but I struggle to understand its relevance. I was hoping the discussion would elaborate on the implications of the results. If one group had a higher mean than another on a variable, why does that matter? The audience will want to understand this. What could be hypothesized to inform the difference? I worry that without situating these results with some discussion on relevance and implications they could be generalized out of context or overlooked as interesting but irrelevant. The final paragraph dives into this a little, but including these insights, even as a brief reflection when first presented would enhance the understanding and views of relevance.

Author Response

Thank you for all your suggestions. You can find all edits in the file I uploaded to the system in red. Kind regards.

Comment 1: Early in the introduction “higher order thinking skills” (used in the title) should be defined and cited before elaborating on the individual sills.

Response 1: “Higher order thinking skills” are now described in the first section of the introduction.

Comment 2: Many claims are made early in the paper about the importance of HOTs and these should be cited as well. This helps the audience understand that the importance of these skills is established beyond the authors opinions.

Response 2: The significance of Hots is emphasized at the beginning of the article.

Comment 3: For Section 1 I would encourage the authors to connect their overview of the skills to current, highly cited literature which creates a common understanding in the field. This will enhance the chances that the field will find it relevant.  Include a commonly used definition of critical thinking and creative thinking if possible and relate those definitions to the chosen scales used for the study.

Response 3: In Chapter 1, the views on skills are supported by references. The description of critical thinking is now included in Chapter 1. The description of creative thinking is now included in Chapter 1.

Comment 4: Line 62, briefly summarize the findings of the 4 studies, it is unclear why they are included.

Response 4: The findings of the studies summarized in line 62 are similar studies identified in the literature to demonstrate the originality of the present study.

Comment 5: Line 119 – Please elaborate on the ‘branches’, it is, for example, unclear what a ‘common’ branch might be, or a ‘skill’ branch. This will help readers to generalize the findings.

Response 5: A detailed discussion on the branches is presented in line 119.

Comment 6: Line 254 -  “the mean critical thinking skill score of the foreign language teachers employed in pre- and primary schools, with 15 years or less experience was higher” – please be explicit about what these scores were ‘higher’ than. I’m not sure if there is a statement missing with the citation [11] and there appears to be some missing information here.

Response 6: Line 254 is edited accordingly.

Comment 7: Line 255 – it is unclear what the reference to Polat is referencing, perhaps this is part of the missing information?

Response 7: Line 255 is edited accordingly.

Comment 8: This study was interesting, but I struggle to understand its relevance. I was hoping the discussion would elaborate on the implications of the results. If one group had a higher mean than another on a variable, why does that matter? The audience will want to understand this. What could be hypothesized to inform the difference? I worry that without situating these results with some discussion on relevance and implications they could be generalized out of context or overlooked as interesting but irrelevant. The final paragraph dives into this a little, but including these insights, even as a brief reflection when first presented would enhance the understanding and views of relevance.

Response: 8: The discussion section is edited accordingly.

Reviewer 2 Report

Comments and Suggestions for Authors

I consider the submitted article to be original. The treated topic is current and its results can be used to improve the quality of the educational process. The correct interpretation of research results can be a contribution to the implementation of changes that will positively affect the subjective experience of teachers and the quality of relationships in school teams.

A richer discussion would increase the quality of the article. In part 4. Discussion, facts from other researches are presented. However, I expect a more accurate comparison of how these researches were different (methodology, number of people studied, country) and how this might have influenced the results of the research. I would also expect the authors to try to suggest possible causes that could have led to the results they measured. These could be the basis of hypotheses for future research.

Author Response

Thank you for all your suggestions. You can find all edits in the file I uploaded to the system in green. Kind regards.

Comment 1: A richer discussion would increase the quality of the article. In part 4. Discussion, facts from other researches are presented. However, I expect a more accurate comparison of how these researches were different (methodology, number of people studied, country) and how this might have influenced the results of the research. I would also expect the authors to try to suggest possible causes that could have led to the results they measured. These could be the basis of hypotheses for future research.

Response 1:

  • Suggested editing is implemented in section 4.
